# Short duration overnight cattle kraaling in natural rangelands: Does time after kraal use affect their utilization by wildlife and above ground grass parameters?

Rangarirai Huruba[1,2,3]*, Servious Nemera[1], Faith Ngute[1,2], Meshack Sahomba[2], Peter J. Mundy[1], Allan Sebata[1], Duncan N. MacFadyen[3]

1 Department of Forest Resources & Wildlife Management, National University of Science & Technology, Ascot, Bulawayo, Zimbabwe, 2 Debshan Ranch, Shangani, Zimbabwe, 3 E Oppenheimer & Son (Pty) Limited, Parktown, South Africa

* rhuruba@gmail.com

## Abstract

In east and southern Africa some private ranch owners are corralling (hereafter kraaling) cattle overnight for short periods (for example, seven days) in natural rangelands to create nutrient enriched hotspots which are attractive to large herbivores. However, the effect of season and time after kraal use (alt. age of nutrient enriched hotspots) on large herbivore use of these sites has not been examined. We collated the number of large herbivore sightings per day from camera traps during wet, early and late dry season in nutrient enriched hotspots of varying ages (1, 2, 3 and 4 years) and surrounding vegetation. In addition, above ground grass biomass and height in nutrient enriched hotspots was compared to that of the surrounding vegetation. Furthermore, we tested if repeated grazing in nutrient enriched hotspots stimulated grass compensatory growth. Large herbivore use of nutrient enriched hotspots was similar during wet, early and late dry season. Time after kraal use had a significant effect on mixed feeders (impala and African savanna elephant) utilization of nutrient enriched hotspots but not grazers (zebra and warthog) and browsers (giraffe and greater kudu). Both impala and African savanna elephants mostly used nutrient enriched hotspots one year after kraal use. Aboveground grass biomass and height were higher in surrounding vegetation than in nutrient enriched hotspots. Repeated clipping (proxy for grazing) resulted in compensatory aboveground grass biomass in nutrient enriched hotspots, which declined with time after kraal use. We concluded that nutrient enriched hotspots created through short duration overnight kraaling were important foraging sites for large herbivores.

## Introduction

In African savanna ecosystems availability of nutritive forage is important for both domestic and wild herbivores. Old cattle bomas or corrals (also referred to as glades) are considered

**Data Availability Statement:** All relevant data files are available from the DANS database (https://doi. org/10.17026/dans-zud-vp6x).

**Funding:** DM and RH received support from E Oppenheimer & Son Pvt Ltd in form of salaries for the period of the study and are still employed under the Research and Conservation division. MS received support from Debshan Pvt Ltd through the form of a salary. The funders had no role in study design, data collection and analysis, decision to publish, or preparation of the manuscript. The specific roles of these authors are articulated in the 'author contributions' section.

**Competing interests:** DM and RH are salaried employees of E Oppenheimer & Son Pvt Ltd, currently employed under the Research and Conservation division. MS is a salaried employee of Debshan Pvt Ltd. There are no patents, products in development or marketed products associated with this research to declare. This does not alter our adherence to PLOS ONE policies on sharing data and materials."

important nutrient hotspots in east and southern Africa [1–5]. Large herbivores forage in old bomas and also make use of their openness to seek refugee against predators [6]. The use of bomas to pen cattle overnight alters the ecosystem functions through redistribution of nutrients within the terrestrial ecosystem [7]. Large herbivores forage from the surrounding landscape and deposit nutrients in bomas as dung and urine overnight [8,9], resulting in translocation of nutrients. Repeated use of old bomas by large herbivores keeps them nutrient (particularly nitrogen and phosphorus) enriched and productive through dung and urine addition [10]. Nitrogen is mostly recycled as urine and phosphorus through dung deposition [11]. Some private ranch owners in east and southern Africa are now corralling (hereafter kraaling) cattle overnight for short periods (for example, seven days) in natural rangelands to create nutrient hotspots (hereafter nutrient enriched hotspots) similar to old bomas [1–4]. Although previous studies have reported preferential use of old bomas by wildlife [12–14], few studies have monitored the use of nutrient enriched hotspots by wildlife (but see [1]).

Large herbivores can be classified into three feeding guilds *viz*. grazers, mixed feeders and browsers [15]. Within these feeding guilds large herbivores show variation in their adaptations to the quality of forage. For instance, Burchell's zebra (*Equus quagga burchelli*) is a large grazer tolerant to fibrous diets, warthog (*Phacochoerus africanus*) is a small grazer intolerant of fibrous diets, impala (*Aepyceros melampus*) is a highly selective medium size mixed feeder, African savanna elephant (*Loxodonta africana africana*) is a less selective large mixed feeder, while giraffe (*Giraffa camelopardalis giraffe*) and greater kudu (*Tragelaphus strepsiceros*) are large obligate browsers [16,17]. Monitoring the use of nutrient enriched hotspots by large herbivores in the three feeding guilds is important in understanding the impacts of manipulating rangelands through short duration overnight cattle kraaling.

The use of nutrient enriched hotspots by large herbivores in African savanna ecosystems that are characterized by distinct seasonality (wet and dry season) is expected to vary with season. Generally, grass is green and nutritious during wet season, but brown and less nutritious during dry season, influencing large herbivore foraging decisions. However, in nutrient enriched hotspots repeated grazing stimulates grass resprouting even in the dry season if there is adequate soil moisture. Thus, the differences in nutritive value of grass between wet and dry season is expected to influence use of nutrient enriched hotspots. For instance, large herbivores are expected to use nutrient enriched hotspots and surrounding landscape similarly during the wet season but to predominantly use the former during the dry season. Mayengo et al. [18] observed that grass resprouting resulted in preferential use of grazing lawns during the dry season. In southern Africa the wet season occurs between November and April, while the dry season is between May and October.

Soil nutrients in nutrient enriched hotspots decline with time after kraal use [19]. For example, soil nutrients are highest within twenty-four months of kraal use and thereafter decline (Huruba unpublished data). The loss of soil nutrients with time after kraal use results in a decrease in grass quality leading to a decline in the use of nutrient enriched hotspots by grazers [1–3]. Grass nutrient quality influences the selection of foraging patches by grazers [20]. Thus, grazer use of nutrient enriched hotspots is expected to be high within two years of kraal use and thereafter decline with time. However, a positive grass-herbivore feedback loop and continued nutrient deposition through dung and urine could maintain high grass nutrient content long after kraal use. The effect of time after kraal use on the utilization of nutrient enriched hotspots by large herbivores needs to be investigated to better understand the benefits of this practice of rangeland manipulation.

Above ground grass biomass and height in nutrient enriched hotspots is regulated by two factors, *viz*. soil nutrient content and large herbivore grazing. Grass responds to improved soil fertility through rapid growth. However, attraction of grazers to nutrient enriched hotspots, in

response to availability of abundant and nutritive grass [21,22], is expected to result in intense grazing, leading to reduced aboveground grass biomass and height. The highest above ground grass biomass in nutrient enriched hotspots is recorded just after the first rains post kraaling as soil nutrients are at their peak, and thereafter decline with time after kraal use [2]. However, grazing intensity also regulates above ground grass biomass and height. In order to determine the effect of age of nutrient enriched hotspots on above ground grass biomass and height the amount of grass cropped by large herbivores needs to be measured. Impalas select short, low fiber grass which is highly digestible [22], while zebras select foraging patches with high above-ground grass biomass to achieve high digestive fill, because as hindgut fermenters they have a fast digesta passage rate [23]. However, zebra and other equids consume both short and tall grass to balance between forage quality and quantity [24]. For example, zebra have narrow muzzle considered well suited for clipping tall grasses [25]. Grass height influences herbivore habitat use [26].

Repeated grazing creates a positive herbivore-grass feedback loop that maintains high plant nutrient levels [27]. Grass resprouts are rich in nutrients and result in repeated grazing [3,28]. Nutrient enriched hotspots are subjected to repeated grazing which in the long term could result in grass failing to compensate lost biomass [29]. The ability of grass to compensate lost biomass under repeated grazing needs to be determined to better understand the effect of creating nutrient enriched hotspots in rangelands. Although repeated grazing is stimulatory to growth [30], grass responds through under-, partial- or over-compensation of lost foliar tissue [28]. Generally, most grass either under or equally- compensate the lost biomass because grazing results in loss of photosynthetic material, limiting the ability of the plants to photosynthesize [31]. Grass regrowth in nutrient enriched hotspots benefit from enhanced soil fertility due to dung and urine deposition [32,33]. In this study we investigated if repeated grazing resulted in compensatory aboveground grass biomass in nutrient enriched hotspots of different ages.

The use of nutrient enriched hotspots by large herbivores can be studied using camera traps as they are cost-effective, efficient and non-intrusive [12–14]. Camera traps can be used to determine spatial and temporal use of foraging resources by large herbivores [16]. For example, Young et al. [32] used camera traps to study forage selection by grazers in Kruger National Park. The use of foraging resources by large herbivores can be influenced by their abundance. Hence, large herbivore abundance need to be related to the number of sightings in camera traps to calculate relative abundance indices to ascertain if the use of nutrient enriched hotspots is affected by population. The probability of sighting animals in camera traps is strongly influenced by their population [29].

We studied the use of nutrient enriched hotspots of varying ages (1, 2, 3 and 4 years after kraal use) by six large herbivores of different sizes (zebra, warthog, impala, African savanna elephants, giraffe and greater kudu) and feeding guilds (grazers–zebra and warthog; mixed feeders–impala and African savanna elephant; and browsers–giraffe and greater kudu) during different seasons (wet, early dry and late dry season). We collated the number of large herbivore sightings in nutrient enriched hotspots from camera trap photographs during the wet (January), early dry (June) and late dry (October) season. In addition, above ground grass biomass and height were measured and related to grazer sightings. Furthermore, we carried out an experiment on simulated grazing to determine the response of grass growth to repeated (three times during growth season) clipping. We tested the following hypotheses: 1) season affects use of nutrient enriched hotspots by large herbivores, 2) age of nutrient enriched hotspots affects its use by large herbivores, 3) aboveground grass biomass and height varies with time after kraal use 4) repeated grazing result in compensatory aboveground grass biomass. This study was conducted at Debshan, a mixed cattle-wildlife ranch, located in central Zimbabwe.

## Materials & method

### Study site

Debshan ranch is located in central Zimbabwe (29°13′E, 19°36′S; 1230m elevation) (Fig 1). It is a mixed cattle-wildlife ranch that covers an area of 800 km² and supports a diversity of large mammal species that include impala, Burchell's zebra, warthog, African savanna elephant, northern giraffe and greater kudu. The study area is characterized by a catenal vegetation pattern, with most areas consisting of grassed bushland with patches of Miombo woodland [34]. The dominant woody species is *Acacia karroo* Hayne with the major grass species being *Hyparrhenia filipendula* (Hochst.) Stapf., *Eragrostis curvula* (Schrad.) Nees., *Heteropogon contortus* (L.) Roem. & Schult., *Bothriochloa insculpta* (Hochst. Ex A. Rich.), *Digitaria milanjiana* (Rendle) Stapf., and *Panicum maximum* Jacq. Mean annual rainfall is 612 mm, with a rainy season that runs from November to March and a dry season from April to October [34]. Mean annual temperature is 22.6°C, with October (31.4°C) the hottest month and July the coldest (8.5°C).

### Creation of nutrient enriched hotspots

The short duration overnight cattle kraaling system which created the nutrient enriched hotspots (alt. previously kraaled sites) was introduced to Debshan ranch in 2012. A herd of cattle (approximately 400) is kept overnight in a kraal (70m by 100m) set up in the rangelands for seven days before being moved to a new location. The kraal is made of heavy, steel posts and thick canvas sheeting and is 1.5m high [2]. Metal poles and wire cable are used to keep the boma canvas sheeting in place. Sites for placement of kraals are randomly selected and no trees are cut. The minimum distance between any two kraals is approximately two km. The cattle are guarded by herders who sleep in portable houses adjacent to each kraal overnight to protect them from predators. During daylight hours, herders direct cattle to grazing areas and drinking points close to the kraals. The kraals are used once for a period of seven days and have water supplied in a trough.

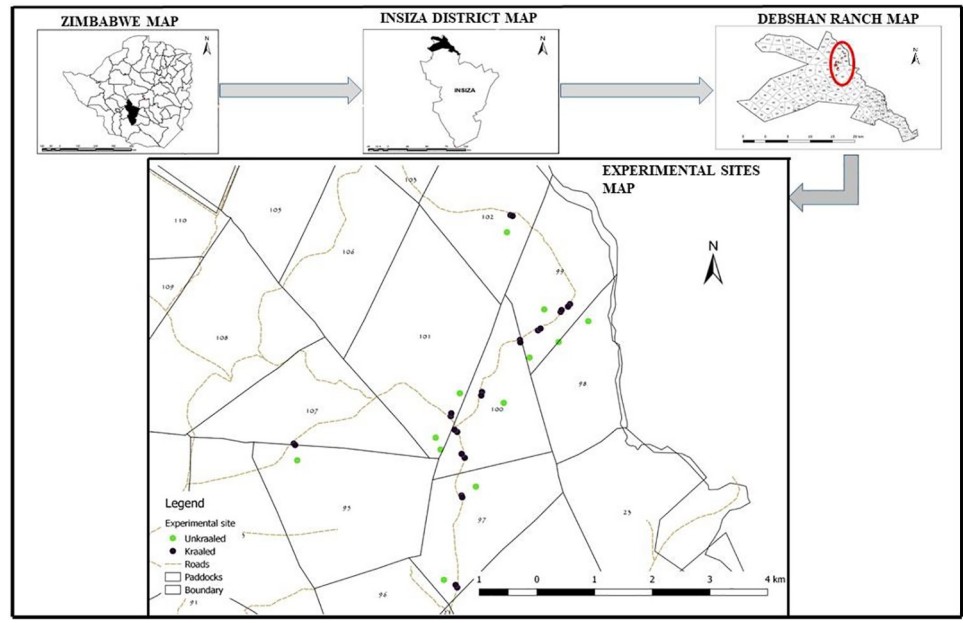

**Fig 1. Debshan ranch location showing camera trap positions.**

## Camera traps setting

We deployed Cuddeback Attack/Attack IR digital scouting cameras ($n = 11$), Cuddeback C (modular) and E model cameras ($n = 25$) (Cuddeback Trail Camera company, India) infrared camera traps at eighteen locations in nutrient enriched hotspots (alt. previously kraaled sites) of varying ages (1, 2, 3 and 4 years) and control sites in surrounding vegetation at Debshan ranch between January and October 2017 (Fig 1). Three kraals were randomly selected for each year treatment with each having a control marked 300m away. Each kraal replicate and control site had two cameras. The cameras were mounted on tree trunks at one meter above the ground to detect medium- to large-bodied mammals [35]. Cameras were set for pictorial (single capture/minute) data capture for diurnal and nocturnal animal at a trigger speed interval of 60 seconds and each image displayed date (dd/mm/yy), time (hh:mm) and camera number(ID). Secure Digital (SD) memory cards and non-rechargeable batteries were replaced at two week interval. The pictorial data was downloaded from the SD cards and stored in folders labeled according to kraal age. Microsoft Excel version 2016 was used to store the photographic data with the following details: camera location (kraaled or unkraaled area), camera unit identifier, date (dd/mm/yy), time (hours, minutes) and animal species. Data collection was done in January ($n = 30$ days), June ($n = 30$ days) and October ($n = 30$ days) 2017. The number of animal sightings of each large herbivore species during each period was recorded from the camera trap data and expressed as number of animal sightings per day. All successive photographs of a species at the same camera were treated as independent if ten minutes passed with no captures of the particular species [36]. Camera trapping is non-intrusive and effective in studying large herbivore spatial and temporal use of habitats [37].

## Aerial census data

Aerial censuses are conducted annually at Debshan ranch using a helicopter during July-August. Data for the year 2017 is presented in this study and used to calculate sighting indices for the six large herbivores studied.

## Sighting indices

Sighting indices were calculated using the formula:

$$\text{Sighting index} = \text{Number of wildlife sightings}/\text{Total wildlife population}$$

## Estimates of aboveground grass biomass cropped by grazers

We set up four chicken wire mesh (2 cm diameter holes) herbivore exclusion movable cages (1m × 1m × 1 m) in each nutrient enriched hotspot (alt. previously kraaled site) and surrounding vegetation to estimate aboveground grass biomass cropped by grazing herbivores. The cages were kept in the same position during the growth season (October 2016 to May 2017). The difference in aboveground grass biomass inside and outside the movable cages was assumed to represent grass cropped by the grazing herbivores [38]. Aboveground grass biomass both inside and outside the movable cages was clipped using a clipper, air dried, before oven drying at 60°C for 48 hours and then weighed. All the aboveground grass inside the movable cage was clipped to ground level. Cropped aboveground grass biomass was then calculated as the difference between aboveground grass biomass inside and outside the mobile cages. Grass height was also measured in each sampling site using a tape measure to the nearest mm.

To test if repeated grazing results in compensatory aboveground grass biomass we clipped grass inside movable cages three times during the growth season and compared with aboveground grass biomass in a single clipping at the end of the growth season. Grass was clipped to

ground level on each occasion. Aboveground grass biomass inside movable cages located in nutrient enriched hotspots (alt. previously kraaled sites) of different ages (1, 2, 3 and 4 years after kraal use) were clipped three times at twenty-eight day intervals from the beginning of the growth season (28/01/2017) (1st clipping), peak of the growth season (25/02/2017) (2nd clipping) and at the end of the growth season (25/03/2017) (3rd clipping). The aboveground grass biomass removed at each of the three clippings was recorded and the sum for all the clipping calculated. Compensatory aboveground grass biomass ($gm^{-2}$) was calculated using the formula: total aboveground grass biomass for all three clippings–aboveground grass biomass clipped once at the end of growth season.

## Statistical analysis

A total of 2833 camera images captured during the study period (90 days) were used for analysis of number of wildlife sightings [wet (January): $n = 324$; early dry (June): $n = 874$; late dry season (October): $n = 1635$]. The wildlife sightings were expressed as number of sightings per day to allow comparison of data for the three periods (wet, early dry and late dry season) as the number of camera trap images varied with season.

All data were tested for homogeneity of variance and normality using Levene statistics and Shapiro-Wilk test, respectively prior to statistical analyses. The effect of season (wet, early dry and late dry) and age of kraal (no kraaling, one, two, three and four years after kraaling) on number of wildlife sightings per day (proxy for use of nutrient enriched hotspots) were tested using one way analysis of variance. Above ground grass biomass, cropped grass biomass, clipped grass biomass and compensatory grass growth among the different aged kraal sites (no kraaling, one, two, three and four years after kraaling) were also compared using one way analysis of variance. The sighting index for the six wildlife species (zebra, warthog, impala, African savanna elephants, giraffe and greater kudu) were compared using one way analysis of variance. Where differences among treatments were significant Tukey's HSD was used for pairwise *post hoc* comparisons. Burchell's zebra and warthog (grazers) number of sightings per day were related to aboveground grass biomass and height using Pearson square correlation. Sighting index was also related to wildlife population using Pearson square correlation. All data analysis was carried out using IBM SPSS 16.

## Results

Zebra, warthog, impala, African savanna elephants, giraffe and greater kudu had large number of sightings per day in the camera traps to be used as representative species for the three feeding guilds (grazers, mixed feeders and browsers). Zebra and warthog are grazers, impala and elephants mixed feeders and giraffe and greater kudu obligate browsers. Other wildlife species with low numbers sighted in the camera traps were hare (*Lepus capensis*), common duiker (*Sylvicapra grimmia*), and steenbuck (*Raphicerus campestris*).

For each of the six species, there were no significant differences in sightings across the three seasons (early, early dry and late dry) (Burchell's zebra: $F_{2,9} = 1.45$, $p = 0.285$; warthog: $F_{2,9} = 2.40$, $p = 0.146$; impala: $F_{2,9} = 0.72$, $p = 0.511$; African savanna elephant: $F_{2,9} = 0.02$, $p = 0.899$; giraffe: $F_{2,9} = 0.08$, $p = 0.926$; greater kudu: $F_{2,9} = 0.24$, $p = 0.795$) (Fig 2). Impala ($F_{4,10} = 11.06$, $p = 0.001$) and African savanna elephant ($F_{4,10} = 153.45$, $p < 0.001$) sightings per day significantly varied with time after kraal use (alt. age of nutrient enriched hotspots), while there were no significant differences for Burchell's zebra ($F_{4,10} = 2.80$, $p = 0.09$), warthog ($F_{4,10} = 1.66$, $p = 0.24$), giraffe ($F_{4,10} = 2.11$, $p = 0.15$) and greater kudu ($F_{4,10} = 2.39$, $p = 0.12$) (Fig 3). Impala mostly used nutrient enriched hotspots one and four years after use, while African savanna elephant mostly preferred to use nutrient enriched hotspots one year after kraaling.

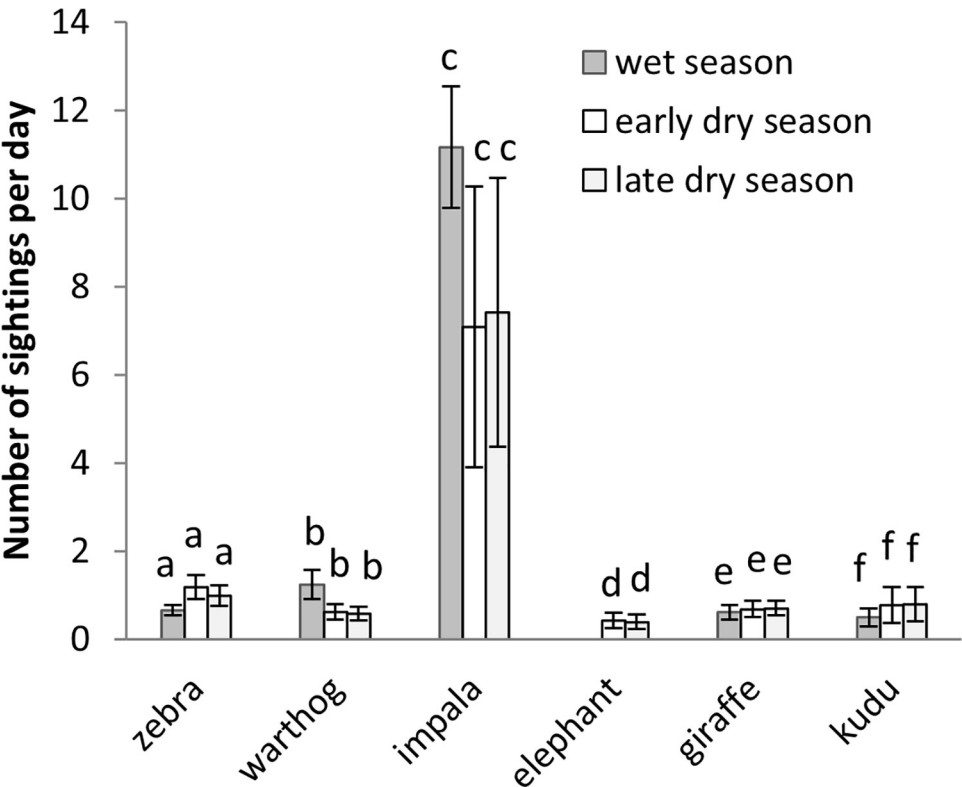

**Fig 2. Mean (±SE) number of wildlife sightings per day during three periods (wet, early dry and late dry season) in nutrient enriched hotspots.** Similar letters show that there were no significant differences among the treatments.

Aboveground grass biomass was highest in surrounding vegetation (unkraaled sites) ($F_{4,20}$ = 1167, $p < 0.001$), with most aboveground grass biomass cropping occurring in the three year old nutrient enriched hotspots ($F_{4,20}$ = 112.98, $p < 0.001$) (Fig 4). Grass was tallest in surrounding vegetation (unkraaled sites) ($F_{4,20}$ = 407.13, $p < 0.001$; Fig 5). Zebra and warthog sightings were not significantly correlated to aboveground grass biomass (zebra: $r$ = 0.54, $p$ = 0.34, $n$ = 5; warthog: $r$ = - 0.68, $p$ = 0.21, $n$ = 5) and grass height (zebra: $r$ = 0.57, $p$ = 0.31, $n$ = 5; warthog: $r$ = - 0.76, $p$ = 0.14, $n$ = 5). Aerial census counts at Debshan ranch in 2017 showed that impala and giraffe were the most and least abundant wildlife species respectively (Fig 6). Giraffe had the highest sighting index ($F_{5,18}$ = 7.02, $p$ = 0.001; Fig 7). Sighting index was not significantly correlated to wildlife population ($r$ = 0.10, $p$ = 0.85, $n$ = 6). Repeated grass clipping (proxy for grazing) resulted in compensatory above ground grass biomass, with the highest in the one year after kraal use sites (Table 1).

## Discussion

Our research highlights the importance of using short duration overnight cattle kraaling in rangelands to create nutrient enriched hotspots attractive to wildlife in an African savanna ecosystem. We used the number of animal sightings per day from camera traps as a proxy for use of nutrient enriched hotspots and surrounding vegetation. All the six large herbivores (zebra, warthog, impala, African savanna elephants, giraffe and greater kudu) used nutrient enriched hotspots throughout the year. Previous studies have reported impala, warthog, African savanna elephants and other large herbivores as using nutrient enriched hotspots [1,3,5,12].

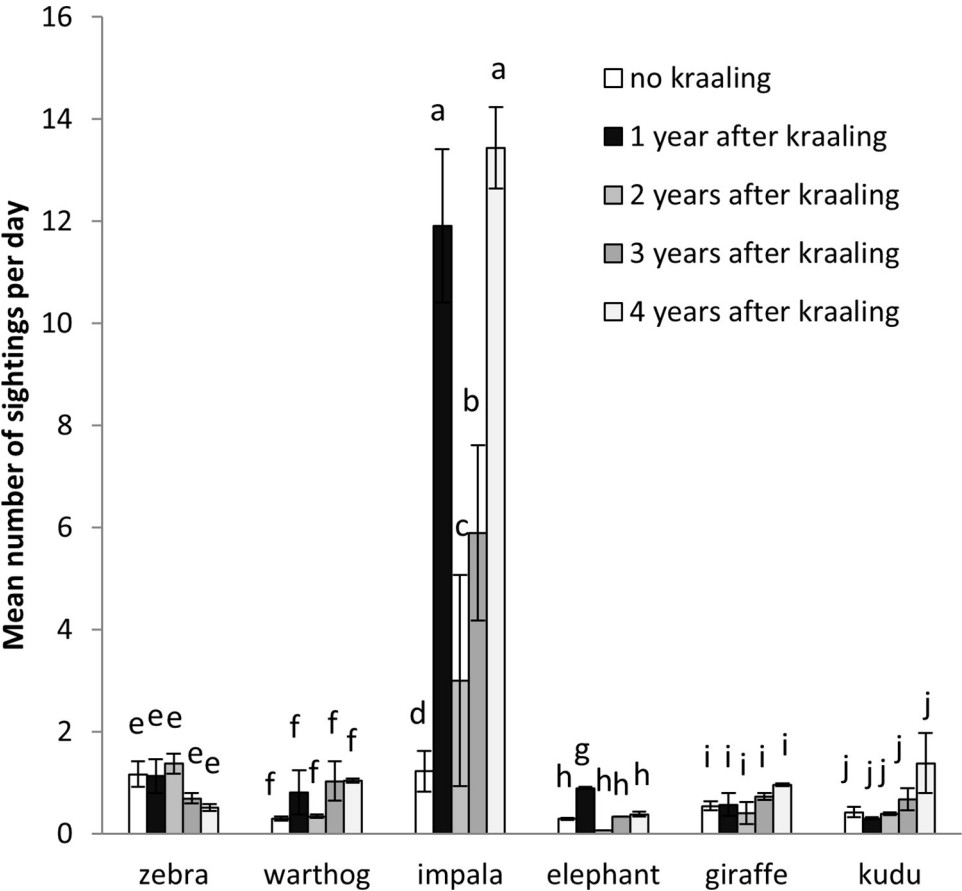

**Fig 3. Mean (±SE) number of wildlife sightings per day in nutrient enriched hotspots of varying ages.** Similar letters show that there were no significant differences among the treatments.

Our findings did not support the first hypothesis as wildlife use of nutrient enriched hotspots was similar during the wet and dry season. This suggests that the nutrient enriched hotspots provided nutritive forage and / or refugee to wildlife during both wet and dry season [1,39].

Our results showed that giraffe used nutrient enriched hotspots more than the surrounding vegetation. Conversely, Veblen and Porensky [5] reported giraffes as avoiding nutrient enriched hotspots, instead foraging in the surrounding vegetation. Our short duration overnight cattle kraaling system did not cut down trees and shrubs, with plant damage only due to cattle trampling. Traditional glades in east Africa are treeless because trees are cut down for use as kraal fences [40]. Zebra, African savanna elephants and kudu had low sighting indices, implying low use of nutrient enriched hotspots. Veblen and Porensky [5] also reported zebra as not actively seeking out high quality forage in nutrient enriched hotspots, presumably, because their large size and hind-gut fermentation allowed them to consume fibrous diets. In addition, low aboveground grass biomass in nutrient enriched hotspots, presumably, made them less attractive to zebra [4].

Our results showed that only mixed feeders use of nutrient enriched hotspot varied with time after kraal use (alt. age of previously kraaled sites). For both impala and African savanna elephant nutrient enriched hotspots were mostly used one year after kraal use. This was, presumably, due to the openness of these sites which improved impala predator detection and

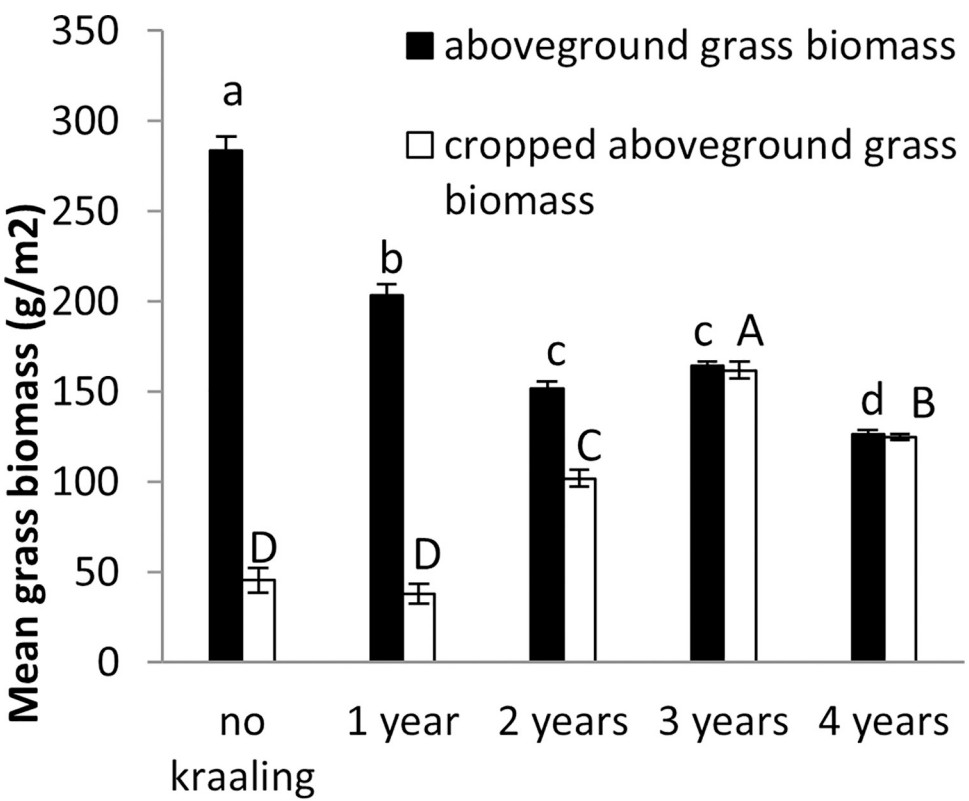

**Fig 4. Mean (±SE) aboveground and cropped grass biomass at nutrient enriched hotspots of different ages.** Different letters (a, b, c and d—for aboveground grass biomass; and A, B, C, D—for cropped aboveground grass biomass) show differences in the treatments.

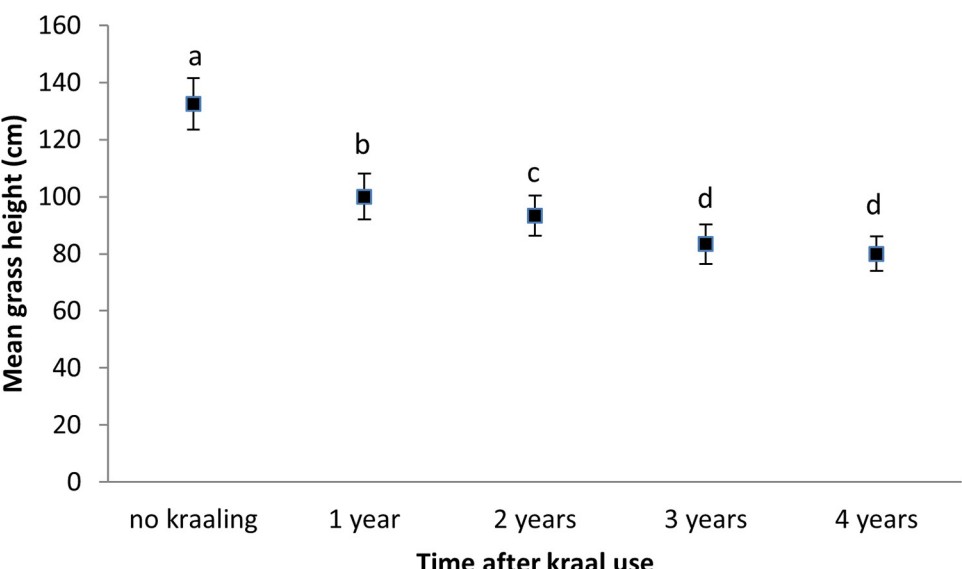

**Fig 5. Mean (±SE) grass height in nutrient enriched hotspots of different ages.** Similar letters show that there were no significant differences between the treatments.

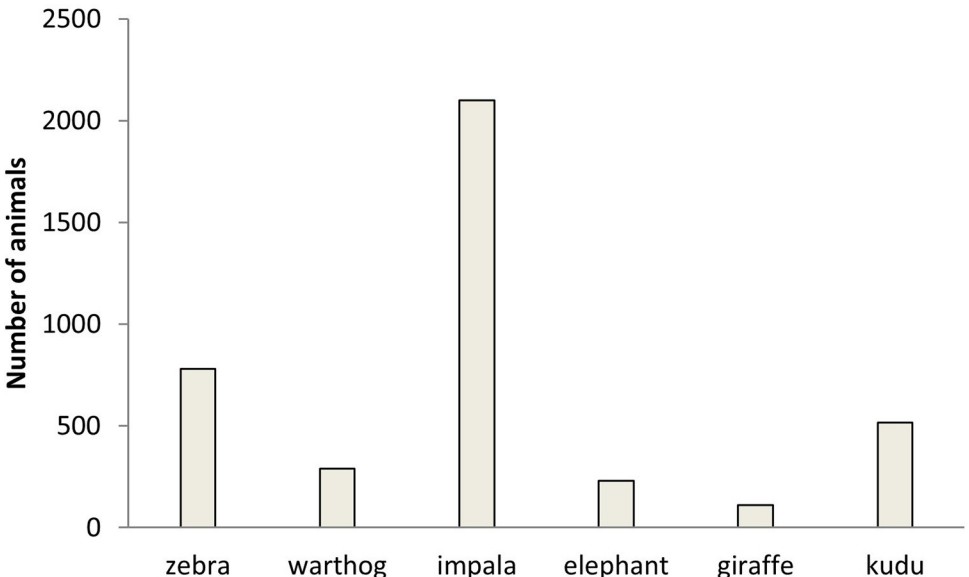

**Fig 6. Number of wildlife at Debshan ranch in 2017.**

allowed African savanna elephant easy movement. No plausible explanation could be proffered for the impala use of four year old nutrient enriched hotspots. Previous studies reported nutrient enriched hotspots as most attractive to mixed feeders [14,41,42]. Shannon et al. [42] attributed this to the ability of mixed feeders (particularly African savanna elephants) to mainly browse while also consuming grass. Huruba et al. [1] reported cattle as breaking woody plant stems and stripping them of foliage during overnight kraaling initiating resprouting, with the resprouts attractive to impala due to high foliar nitrogen and low condensed tannin concentrations.

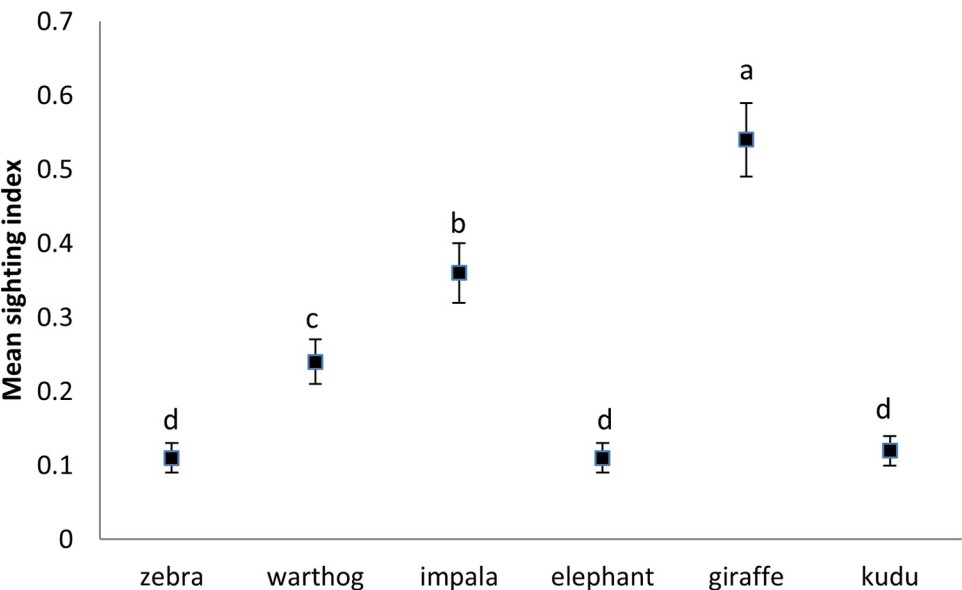

**Fig 7. Mean (±SE) sighting index for six wildlife species.** Similar letters show that there were no significant differences between the treatments.

**Table 1. Mean (±SE) aboveground grass biomass (gm⁻²) in cages clipped once and repeatedly (three times) in previously kraaled sites of different ages.**

| | Clipped once | Repeated clipping | | | | Compensatory growth |
| --- | --- | --- | --- | --- | --- | --- |
| | | 1st clipping | 2nd clipping | 3rd clipping | Total for repeated clipping | |
| 1 year after kraal use | 248.40[b] ± 4.46 | 379.20[a] ± 2.85 | 133.80[b] ± 1.69 | 85.20[a] ± 1.77 | 599.20[a] ± 2.08 | 351.20[a] ± 3.31 |
| 2 years after kraal use | 252.40[b] ± 3.75 | 331.20[b] ± 15.99 | 155.40[a] ± 1.29 | 72.60[b] ± 0.93 | 535.20[b] ± 1.83 | 274.20[b] ± 1.66 |
| 3 years after kraal use | 287.40[a] ± 3.33 | 281.60[c] ± 2.20 | 131.60[b] ± 1.62 | 71.05[b] ± 2.52 | 497.40[c] ± 1.81 | 209.40[c] ± 2.58 |
| 4 years after kraal use | 255.20[b] ± 4.19 | 205.00[d] ± 1.61 | 112.00[c] ± 0.95 | 37.00[c] ± 2.10 | 353.00[d] ± 4.36 | 101.00[d] ± 1.58 |
| | $F_{3,16} = 20.54$, $p < 0.001$ | $F_{3,16} = 81.55$, $p < 0.001$ | $F_{3,16} = 152.58$, $p < 0.001$ | $F_{3,16} = 116.88$, $p < 0.001$ | $F_{3,16} = 1453.00$, $p < 0.001$ | $F_{3,16} = 1964.00$, $p < 0.001$ |

Aboveground grass biomass and height were highest in surrounding vegetation, presumably, because of low grazing intensity. This is supported by the low aboveground grass biomass cropping in the surrounding vegetation. These results show that nutrient enriched hotspots were more intensely grazed than surrounding vegetation. Huruba et al. [2] reported warthogs as intensely grazing in nutrient enriched hotspots. Interestingly, aboveground grass biomass was not significantly correlated to zebra and warthog number of sightings per day. Generally, aboveground grass biomass and height in nutrient enriched hotspots tended to decrease with time after kraal use (see Fig 4), with cropping showing an opposite trend. Aboveground grass biomass cropped was within the range of 89 to 951 gm⁻² reported in the Kruger National Park by Burkepile et al. [43]. Grass in the surrounding vegetation was moribund and unattractive even to zebra that are tolerant to fibrous diets. Zebra are tolerant to fibrous low quality grass because of their fast passage rate of forage through the gastrointestinal tract [44].

Repeated clipping (proxy for grazing) resulted in compensatory aboveground grass biomass in nutrient enriched hotspots. McNaughton [27] reported grasses in the Serengeti as overcompensating lost foliage. Compensatory aboveground grass biomass declined with time after kraal use, presumably, due to a decline in soil fertility. Improved soil fertility due to dung and urine deposition, particularly one year after kraal use, could have enhanced grass compensatory growth. Venter et al. [45] reported nutrient addition (in the form of animal dung) as increasing aboveground grass biomass. The decline in aboveground grass biomass regrowth between first and third clipping (see Table 1) was, presumably, due to resource exhaustion as a result of multiple grass resprouting in response to clipping [46]. Mudongo et al. [47] also reported a decrease in aboveground grass biomass regrowth with increasing clipping frequency. In the long-term repeated grazing could negatively affect tillering leading to the loss of the grass [48]. While previous studies have shown that grazing in the preceding growth season reduces grass productivity in the next growth season [49,50], our results show that a decline in regrowth occurs in the current season. In addition, the decline in aboveground grass biomass regrowth with repeated clipping could be attributed to reduced soil moisture availability with advancing growth season which negatively affects nutrient mineralization [51]. Soil mineralization is higher early in the growth season when soil moisture is high as compared to late in the dry season [47].

The purpose of calculating sighting index was to determine if use of nutrient enriched hotspots was influenced by wildlife abundance. Thus, the fact that sighting index was not significantly correlated to wildlife population implies that use of nutrient enriched hotspots was independent of animal abundance. For example, giraffe had the lowest population and highest sighting index, implying that there were frequent users of nutrient enriched hotspots.

## Conclusion

Our findings showed that nutrient enriched hotspots created through short duration overnight cattle kraaling in natural rangelands were attractive to large herbivores. Large herbivore use of

nutrient enriched hotspots was similar during wet and dry season. Time after kraal use (alt. age of previously kraaled sites) had an effect on mixed feeders (impala and African savanna elephants) use of nutrient enriched hotspots but not grazers (zebra and warthog) and browsers (giraffe and greater kudu). Aboveground grass biomass and height in nutrient enriched hotspots was lower than in surrounding vegetation due to more intense grazing by large herbivores. Repeated grazing resulted in compensatory grass growth that declined with age of nutrient enriched hotspots. The creation of nutrient enriched hotspots in rangelands improves the availability of foraging resources to large herbivores in both wet and dry season.

## Acknowledgments

We also wish to thank the associate editor and two anonymous reviewers for constructive and insightful contributions to this manuscript.

## Author Contributions

**Conceptualization:** Rangarirai Huruba, Peter J. Mundy, Allan Sebata, Duncan N. MacFadyen.

**Data curation:** Rangarirai Huruba, Servious Nemera, Faith Ngute, Meshack Sahomba.

**Formal analysis:** Peter J. Mundy, Allan Sebata.

**Funding acquisition:** Rangarirai Huruba.

**Investigation:** Faith Ngute, Peter J. Mundy, Allan Sebata.

**Methodology:** Rangarirai Huruba, Meshack Sahomba, Peter J. Mundy, Allan Sebata.

**Project administration:** Rangarirai Huruba, Allan Sebata.

**Resources:** Rangarirai Huruba, Duncan N. MacFadyen.

**Supervision:** Rangarirai Huruba, Peter J. Mundy, Allan Sebata.

**Validation:** Allan Sebata.

**Writing – original draft:** Rangarirai Huruba.

**Writing – review & editing:** Peter J. Mundy, Allan Sebata, Duncan N. MacFadyen.

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
