## [Decision Letter · Decision Letter 0]

26 Apr 2021

PONE-D-21-06800

Short duration overnight cattle kraaling in natural rangelands: does time after kraal use affect their utilization by wildlife and above ground grass parameters?

PLOS ONE

Dear Dr. Huruba,

Thank you for submitting your manuscript to PLOS ONE. After careful consideration, we feel that it has merit but does not fully meet PLOS ONE’s publication criteria as it currently stands. Therefore, we invite you to submit a revised version of the manuscript that addresses the points raised during the review process.

The reviewers provide important feedback on this study, in particular, on clarifying the framing of the study, the details of the methodology used, and interpretation of results. I agree with the reviewers that the introduction needs some reorganization and streamlining, and that more detail is needed regarding the kraaling sites studied and the methods used in the study.

We look forward to receiving your revised manuscript.

Kind regards,

Wendy C. Turner

Academic Editor

PLOS ONE

Journal Requirements:

'The authors received no specific funding for this work.'

We note that one or more of the authors are employed by a commercial company: E Oppenheimer & Son (Pty) Limited

4. We note that Figure 1 in your submission contains map images which may be copyrighted.

We require you to either (a) present written permission from the copyright holder to publish this figure specifically under the CC BY 4.0 license, or (b) remove the figure from your submission:

b. If you are unable to obtain permission from the original copyright holder to publish this figure under the CC BY 4.0 license or if the copyright holder’s requirements are incompatible with the CC BY 4.0 license, please either i) remove the figure or ii) supply a replacement figure that complies with the CC BY 4.0 license. Please check copyright information on all replacement figures and update the figure caption with source information. If applicable, please specify in the figure caption text when a figure is similar but not identical to the original image and is therefore for illustrative purposes only.

5. Please include a copy of Table 3 which you refer to in your text on page 15.

Reviewers' comments:

Reviewer's Responses to Questions

**Comments to the Author**

1. Is the manuscript technically sound, and do the data support the conclusions?

Reviewer #1: Partly

Reviewer #2: Partly

2. Has the statistical analysis been performed appropriately and rigorously? 

Reviewer #1: Yes

Reviewer #2: Yes

3. Have the authors made all data underlying the findings in their manuscript fully available?

Reviewer #1: No

Reviewer #2: Yes

4. Is the manuscript presented in an intelligible fashion and written in standard English?

Reviewer #1: Yes

Reviewer #2: Yes

5. Review Comments to the Author

Reviewer #1: I appreciate the opportunity to review this manuscript. The work examines wildlife visitation of abandoned cattle Kralls spanning from one- to four-year-old and also reports on above grass biomass accumulation in these heavily grazed sites. This work is of interest to a broad audience and underpins some of the benefits associated with both current and legacy effects of livestock wildlife integration in rangelands. The manuscript is fairly well written but may require some major revision before it gets to a level that is acceptable for publication. Particularly, I found the introduction not quite well done, and there are also some fundamental concerns on the design of the study. I have itemized some of these concerns below;

General Comments

First, the introduction section lacks of coherent flow between paragraphs. For example, I found it quite distracting switching from paragraph one (which talks about nutrient hotspots created by old cattle bomas) to paragraph 2 (which talks about effectiveness of camera trapping…. something that I actually don’t think should be presented here) to paragraph 3 (which revisits the same issues in paragraph 1) and later in paragraph 7 camera traps are revisited. Additionally, there is a lot of repetition and contradicting statements from one paragraph to another. I have pointed out some of these under specific comments below. Suffices to say, it is hard to read and follow though the introduction section as presented.

Secondly and most critical, there is little or no information relating to how the cattle bomas were set up. The authors present information about the age of the various bomas; which in real sense doesn’t tell much. For example, we know the trajectory of an abandoned boma depends on many factors including; i) the amount of time it was occupied, ii) what livestock species occupied it, iii) how many individual animals were present, iv) the prevailing conditions during occupation (e.g. was is rainy or dry season), v) prevailing conditions after abandonment—which affects dung decomposition and recolonization dynamics, vi) soil type. All, the factors will affect how utilization patterns of an abandoned boma will unfold. Additionally, only get to learn under the discussion session, that creation of bomas that were used for this study did not involve cutting of trees. This is a novel aspect of this work, since most of abandoned bomas elsewhere are devoid of trees; which affects perceived predation risk.

Thirdly, it is misleading to state that “few studies have monitored the use of nutrient hotspots created through kraaling cattle in temporary mobile overnight kraals by wildlife”. Indeed, a lot of work has been done on this this subject, e.g see the citations below. Instead, the authors should clarify the novel aspects of their work.

Augustine, David J., Veblen, Kari E., Goheen, Jacob R., Riginos, Corinna, and Young, Truman P. 2011. "Pathways for Positive Cattle–Wildlife Interactions in Semiarid Rangelands." Smithsonian Contributions to Zoology. 55–71. https://doi.org/10.5479/si.00810282.632.55

Veblen, K. (2012). Savanna glade hotspots: Plant community development and synergy with large herbivores. Journal of Arid Environments, 78, 119–127.

Donihue, C. M., Porensky, L. M., Foufopoulos, J., Riginos, C., & Pringle, R. M. (2013). Glade cascades: Indirect legacy effects of pastoralism enhance the abundance and spatial structuring of arboreal fauna. Ecology, 94(4), 827–837.

Fourthly, it is not clear how successive photographs of an individual animal or a group of animals were treated. With camera traps, you often get cases of some animals hanging out on front of a camera trap, consequently triggering the camera multiple times. Where these analyzed as independent detections. If so, how about individuals that could possibly have been in the glade, but not immediately in front of a camera trap, hence fewer photographs of them taken. I know this is a thorny issue, but there are number of recommendations about deciding which of the triggers are independent detections. Some studies have set a threshold of 20min, others 30min, and even 1hr. Whatever threshold is used, it is important to be explicit about it

Lastly, in addition to documenting above grass biomass accumulation and compensatory growth, the authors should also have looked at species composition. Growth strategies and response to grazing vary remarkably across different species and we also know that nutrient hotspots like these promote dominance of distinct herbaceous communities, which may actually change over time.

Specific suggestions:

Ln 16: delete “natural”

Ln 18 (and multiple other places): “determine” has a connotation of causality inference. Replace with something like “examine)

Ln 27: delete “to ascertain their use of these nutrient hotspots”

Ln 35-35: would have been more interesting if this compensatory growth in the hotspots was compared with the matrix

Ln 35: “Impala benefited the most….”: The fact that they used the hotspots more frequently doesn’t imply the benefited more. Why not just say: “Impala were attracted to …..”

Ln 38 (and elsewhere): the conclusion about nutrients hotspots increasing rangeland heterogeneity is not supported by the data presented here. How was rangeland heterogeneity measured?

Ln 49: congregate -> are attracted

Ln 53: "lawns, old”: insert coma

Ln 74-74: sentence not clear

Lin 79: there are many studies that report attraction of herbivores to glades. Consider citing some of them here

Ln 96-97: not clear what the ‘feedback loop’ is here

Ln 88-104: sounds repetitive

Ln 124-126: contradicts earlier statement?

Ln 137: “restricts grass height” -> “maintains grass short”

Ln 174-175: This study was conducted in Debshan ranch, located…..

Ln 204-207: hard to understand this sentence

Ln 207-208: What date was recoded here?

Ln 277-278: there is no mention of there this wildlife abundance data is coming from

Ln 330: the explanation as to why impala were attracted to 4yr old glades is not quite convincing: what would occasion the changes in grass and browse?

Reviewer #2: Short duration overnight cattle kraaling in natural rangelands: does time after kraal use affect their utilization by wildlife and above ground grass parameters?

1. The study presents the results of original research

The study investigated the use of previously kraaled sites (aged 1,2,3 and 4 years) by Burchell’s zebra and warthogs (grazers), impala and African savanna elephants (mixed feeders) and northern giraffe and kudu (browsers) during the wet season (January), early dry season (June) and late dry season (October). Camera traps were used to record large herbivore sightings in previously kraaled sites. In addition, an experiment to simulate grazing was carried out to determine the response of grass growth to repeated removal of grass biomass during the 3 stated seasons. The study was carried out at Debshan ranch, a mixed cattle-wildlife ranch in central Zimbabwe. The study presents results of original work.

2. Results reported have not been published elsewhere

To the best of my efforts and checking, I confirm that the results reported in this manuscript have not been published in the peer-reviewed literature. However, it is noted that this manuscript has been deposited in the preprint server bioRxiv. PLOS accepts and supports this practice as per PLOS ONE criteria for publication #2.

3. Experiments, statistics, and other analyses are performed to a high technical standard and are described in sufficient detail.

The study was conducted following the scientific method or process. The data collection methods were described and data analyzed. However, the following needs attention: -

Abstract

Lines 30-32: Please clarify what ‘similarly’ stand for or mean. Results showed that the 8 mammals did not use the previously kraaled sites in the same way. So, what does “similarly; mean?

Introduction

• Lines 68-72 This paragraph is too short and is like a hanging paragraph. Please cite more studies that have used camera traps on nutrient hotspot by wildlife. In addition, state some limitations of using camera traps and how you addressed these in your study. Include, if any, how the results of your studies were or were not affected by the use of camera traps

• Line 16, 90, 91, while it may be obvious, the use of “Short duration overnight” must be clearly defined or elaborated in the study. Did “Short duration overnight” imply that the cattle were at these spots just for one night hence short duration? If yes, why not refer to these only as “Overnight?”. If cattle were kraaled for few nights at a time then should such number of nights just be stated? This needs to be made clear in the text.

• Line 126-27 You refer to previously kraaled sites that are used by grazers that deposit dung and urine as “newly created” nutrient hotspots. Why do you refer to then as newly created? Why not refer to them as “nutrient enriched or recharged previously kraaled sites? I suggest this because authors asserted that kraaling adds nutrients to the soil. So, why does it now become “newly created” when it has been established already by cattle kraaling?

• Line 153-154. You state that “However, the positive herbivore-grass feedback has not been widely tested, particularly in nutrient hotspots”. If authors had stated.. “has not been tested” If would indicate no testing has been done elsewhere before. However, by stating that it has not been widely tested, it suggests that it has been tested but in a limited way. If this is the case, in what limited way has this positive loop been tested? What were the outcomes and how do these inform or influence your study?

Materials and methods

While the materials and methods are well described in the manuscript, there are some sections that need more detailed attention. These are listed below: -

Line 197 Camera traps setting

• Line 199-203: Specify how many camera traps were set at each of the previously kraaled sites aged 1,2,3 and 4 years? How many replicates of the previously kraaled sites by age (1,2,3 and 4 years) were established? How many replicates of the control were established and how many camera traps were placed at control sites?

Line 219 Estimates of aboveground grass biomass cropped by grazers

• Line 225-226 Specify how much grass was clipped? At what height from the ground was grass cut? Did you cut the grass all the way to the ground or what? Please specify.

• Line 229. To what nearest measurement was grass height recorded?

• Line 231 Specify the height to which grass were clipped inside movable cages at each of the growth season? Or how close to the ground were grass clipped during each growing season?

243 Statistical analysis

• Line 244: The number of wildlife sightings varied between the 3 sampling seasons i.e January, n=324), June, n=874 and October, n=1635. It is noted that there was huge variation in number sightings such that the October values were 5 times and 2 times larger than the January and June sightings respectively while the June sightings were about 3 times larger than the January values. To what extent did such variation affect the outcomes of the data analysis comparisons? How did this variance affect / influence the outcomes of the data analysis? Did you take into account of this variation and if so how? Please include such in the text.

Results

• Line 254-255. You state that Zebra, warthog, impala, elephant, giraffe and grater kudu were the most frequently sighted in the camera traps. The camera must have captured pictures of mammals other than the 6 study animals. Please list the other mammals apart from the 6. The ranch must surely have other mammals that came to the previously kraaled sites.

• Line 256-258 Authors write….time after kraal use ‘had no significant’ effect…… .In a similar manner and for consistency authors should write …..while wildlife sightings varied significantly with wildlife species…’ [authors should similarly use the word ‘significant’ or ‘not significant’ throughout the results section and discussion where the statistical outcomes reveal so. This will ensure consistency]

• Line 261. Authors should use early season, early dry season and late dry season instead of months January, June and October.

• Lines 266-280 Authors refer to statistical outcomes for results plotted in Figure 3 (number of sightings) and Figure 4 (Population).

o Authors must clarify or describe how the sighting was converted to population based on camera trap photos. This has not been described in the methods section.

o In line 277, authors must indicate that impala and giraffe were the most and least abundant wildlife species at Debsham ranch as estimated from camera trap pictures in 2016. This is important because Camera trap method has limitations as a method or estimating abundance.

o Authors therefore must state the extent to which population estimates presented in Figure 4 (line 617-618) and accurate based on how many of these mammals are present on the entire Debshan ranch?

o Authors need to indicate limitations of such a method to estimate populations of mammals.

• In the results and discussion authors refer to sighting index including Figure 4 (line 621). Yet this is not explicitly indicated so or referred to as such in the Materials and methods especially under statistical analysis.

• In Figure 2 (line 620), Figure 3 (line 606), Figure 5 (line 621) and Figure 6 (line 631) and Figure 7 (line 637) please indicate / include the word ‘Mean’ on the Y-axis labels.

Discussion

• Line 302-303 Authors should state the mechanism which Bailey et al [48] suggested to assert that herbivores were able to identify patches with forage of varying nutritive value. How do herbivores identify nutritive value? How can we be sure? This important because it would contribute to understanding choices which mammals in the present study may have used to select and visit nutrient rich hotspots of previously used kraal sites

• Line 308—310. Consider using the phrase ‘nutrient enriched hotspots” than ‘newly created nutrient hotspots’ when referring to these previously kraaled sites.

• Line 312-319. You make reference to use of woody plants by giraffes in your study. Why did you not quantify, characterize and compare woody plants at the previously kraaled sites to enable reference to woody plants being preferred and eaten by giraffes?

• Line 329-335.

o While authors explained preference of impala and elephants for previously used kraal sites that were one year and four years old (only impala) , they did not explain lack of preference of use of sites that were 2 and 3 years old. Please suggest an explanation and discuss.

o In line 331 authors state that impala and elephants can switch between grasses and browse depending on their nutritive quality. This in my view is a suggested possibility because the nutritive quality of the forage was not investigated or assessed; it was merely inferred. If this is so, it is inaccurate for authors to then conclude in line 332 that …. Therefore, impala and elephants were able to make choices from previously kraaled sites of varying ages depending on the quality of either grasses or browse. Authors must address this.

• Line 343-347. Authors should address the issue of estimation of population of study mammals based in camera trap data raised earlier. In these 5 lines, authors have merely stated the results but have not dis used or interpreted the results neither have they made any reference to literature. What do these results mean and what contribution do these add to the use of previously kraaled sites by mammals in this study?

• Line 352 and 353 In order to assert / suggest the possibility that high aboveground grass biomass may have high risks of predation from ambush-hunting predators such as lions [52] authors must indicate whether lions are present in the Debshan ranch.

• Line 357-358. Authors must explain why warthogs kept aboveground biomass and grass height low in previously kraaled sites that were 3-4 years old compared to those that were 1 to 2 years old and also in general. Although authors refer to Huruba et al. [3] who reported that warthogs intensely grazed in previously kraaled sites, they did not state why? Elaborate?

• Line 361 reference Burkepile et al. [53] is given the same reference number of [53] in line 363 for Groom and Harris [53] instead of [54] as indicated in the reference section. Please correct.

4. Conclusions are presented in an appropriate fashion and are supported by the data.

Conclusions made are generally supported by the data except Line 385 – 386. It is not accurate to conclude that the study has showed that short duration overnight cattle kraaling in natural rangelands creates nutrient hotspots. The use of ‘nutrient’ is inferred and not an outcome of this study because nutrients at the previously kraaled sites were not measured nor quantified and compared but inferred.

5. The article is presented in an intelligible fashion and is written in standard English

This statement is true for this manuscript.

6. The research meets all applicable standards for the ethics of experimentation and research integrity.

Ethics of Experimentation

Authors did not indicate whether they had obtained appropriate research permits and ethical clearance for this study. If they did then they must cite reference numbers for the respective permits and clearance.

Publication Ethics

Authors have specified their respective contributions to the study (line 397-407)

7. The article adheres to appropriate reporting guidelines and community standards for data availability.

Reporting Guidelines

Results were rigorously reported, as appropriate based on the type of the data collected.

Data Availability

Authors have presented data as expected in a scientific journal. The results of the study did not involve gene sequences etc that are deposited following set standards and practice

6. PLOS authors have the option to publish the peer review history of their article (what does this mean?). If published, this will include your full peer review and any attached files.

Reviewer #1: No

Reviewer #2: No

---

## [Author Response · Author response to Decision Letter 0]

3 Dec 2021

We corrected and addressed comments in the manuscript. Fig1 is a map created by author using freeware software from shapefiles generated from free basemaps provided by surveyor general in the country. we have updated the funding statement in cover letter as well.

---

## [Decision Letter · Decision Letter 1]

31 Jan 2022

PONE-D-21-06800R1Short duration overnight cattle kraaling in natural rangelands: does time after kraal use affect their utilization by wildlife and above ground grass parameters?PLOS ONE

Dear Dr. Huruba,

Thank you for submitting your manuscript to PLOS ONE. After careful consideration, we feel that it has merit but does not fully meet PLOS ONE’s publication criteria as it currently stands. Therefore, we invite you to submit a revised version of the manuscript that addresses the points raised during the review process.

 The reviewer has made note of additional clarification needed about the data analysis and results presented.  Please address these comments. Please submit your revised manuscript by Mar 17 2022 11:59PM. If you will need more time than this to complete your revisions, please reply to this message or contact the journal office at plosone@plos.org. Please include the following items when submitting your revised manuscript:A rebuttal letter that responds to each point raised by the academic editor and reviewer(s). You should upload this letter as a separate file labeled 'Response to Reviewers'.A marked-up copy of your manuscript that highlights changes made to the original version. You should upload this as a separate file labeled 'Revised Manuscript with Track Changes'.An unmarked version of your revised paper without tracked changes. You should upload this as a separate file labeled 'Manuscript'.If applicable, we recommend that you deposit your laboratory protocols in protocols.io to enhance the reproducibility of your results. Protocols.io assigns your protocol its own identifier (DOI) so that it can be cited independently in the future. For instructions see: https://journals.plos.org/plosone/s/submission-guidelines#loc-laboratory-protocols. Additionally, PLOS ONE offers an option for publishing peer-reviewed Lab Protocol articles, which describe protocols hosted on protocols.io. Read more information on sharing protocols at https://plos.org/protocols?utm_medium=editorial-email&utm_source=authorletters&utm_campaign=protocols.

We look forward to receiving your revised manuscript.

Kind regards,

Wendy C. Turner

Academic Editor

PLOS ONE

Journal Requirements:

Reviewers' comments:

Reviewer's Responses to Questions

**Comments to the Author**

1. If the authors have adequately addressed your comments raised in a previous round of review and you feel that this manuscript is now acceptable for publication, you may indicate that here to bypass the “Comments to the Author” section, enter your conflict of interest statement in the “Confidential to Editor” section, and submit your "Accept" recommendation.

Reviewer #1: (No Response)

2. Is the manuscript technically sound, and do the data support the conclusions?

Reviewer #1: Partly

3. Has the statistical analysis been performed appropriately and rigorously? 

Reviewer #1: No

4. Have the authors made all data underlying the findings in their manuscript fully available?

Reviewer #1: Yes

5. Is the manuscript presented in an intelligible fashion and written in standard English?

Reviewer #1: Yes

6. Review Comments to the Author

Reviewer #1: The authors have done fairly good job in incorporating suggestions from earlier reviews, there still remains substantial challenges in how the data was analyzed and how the results are presented.

First, I still don’t think the concern about how camera trap data was analyzed was adequately addressed. In their response latter, the authors state that “similar images were treated as one to avoid double counting” but there is no mention of this in the manuscript. Again, what does “similar” mean? If a camera trap captures two photographs within a period of 60 seconds, in one photograph there are three impalas and in the subsequent photograph there are five impalas; are there ‘similar images’? Essentially, multiple detections during a single short occasion are not likely to be independent and thus may contribute little information or may bias estimates. Most studies set a minimum time threshold for independent detections; e.g. all successive photographs of one individual (or species) taken within 10 mins are treated as a simple detection (photographic event). There is plenty of literature around this topic and I encourage authors to look at look at this

Inferences are made about differences across herbivore guilds, in utilization of kralls. However, there are no results to support this, and the authors do not specify which of the six herbivores belongs to which guild.

Ln 243-244: “general linear model (GLM) univariate” -> “univariate general linear model (GLM)”: again, why use univariate models? A more appropriate approach would have been to fist a model that includes all the variables and interactions

Ln 244 “effects” -> “differences”?

Ln 247: I did not see any post hoc analysis presented anywhere in the results

247-249: why just zebra and warthog?

260-261: I don’t see much value in testing the differences in detections across different wildlife species. However, if the authors choose to retain this, they should at least include multiple comparison tests and state which species were actually different

Ln 262-263: rephrase for clarity e.g “For each of the six species, there were no significant differences in sightings across the three seasons [early (January), early dry (June) and late dry (October)]…..”

Ln 265-267: this should be presented above ln 260-261

The authors need to resolve inconsistent use of scientific and common names in reference to different animals e.g. Ln 257-258, ln 272-273 (and elsewhere): The common practice is to give the two names the first time a species is mentioned, and subsequently refer to on one

Line: 274-279: it is not clear what is being tested here. Rephrase for clarity

Ln 280-284: I find these two statements, just like the most of the other statements under results section, quite confusing.

Ln 285-285: how is this “evidence that most grazing occurred in the three-year-old nutrient enriched hotspots.”?

In 282-283, the authors state that three-year-old kralls has the most biomass (most productive), while in 286-287, they state that one year old Kralls had the tallest grass. How is this possible? I thought biomass was correlated with grass height?

Ln 287: new paragraph?

292-293: I still don’t see what value the value of comparing camera trap and aerial census data

294-295: not clear

7. PLOS authors have the option to publish the peer review history of their article (what does this mean?). If published, this will include your full peer review and any attached files.

Reviewer #1: No

---

## [Author Response · Author response to Decision Letter 1]

10 Mar 2022

the comments have addressed and covered in the response to reviewers letter

---

## [Editor Report · Decision Letter 2]

28 Mar 2022

Short duration overnight cattle kraaling in natural rangelands: does time after kraal use affect their utilization by wildlife and above ground grass parameters?

PONE-D-21-06800R2

Dear Dr. Huruba,

We’re pleased to inform you that your manuscript has been judged scientifically suitable for publication and will be formally accepted for publication once it meets all outstanding technical requirements.

Kind regards,

Wendy C. Turner

Academic Editor

PLOS ONE

---

## [Editor Report · Acceptance letter]

5 Apr 2022

PONE-D-21-06800R2 

Short duration overnight cattle kraaling in natural rangelands: does time after kraal use affect their utilization by wildlife and above ground grass parameters? 

Dear Dr. Huruba:

I'm pleased to inform you that your manuscript has been deemed suitable for publication in PLOS ONE. Congratulations! Your manuscript is now with our production department. 

Kind regards, 

on behalf of

Dr. Wendy C. Turner 

Academic Editor

PLOS ONE